# Population Structure, Growth Characteristics, Resource Dynamics, and Management Strategies of *Schizopygopsis younghusbandi* in Four Tributaries of the Yarlung Zangbo River, Tibet

**DOI:** 10.3390/biology14060707

**Published:** 2025-06-16

**Authors:** Haoxiang Han, Lin Wang, Chi Zhang, Hongchi Li, Bo Ma

**Affiliations:** 1Institute of Fisheries Science, Tibet Academy of Agriculture and Animal Husbandry, Lhasa 850000, China; hanhaoxiang98@163.com (H.H.); zc0891@163.com (C.Z.); lihongchi19940702@163.com (H.L.); 2Heilongjiang River Fisheries Research Institute, Chinese Academy of Fishery Sciences, Harbin 150070, China; 3Chinese Academy of Fishery Science, Beijing 100141, China

**Keywords:** *Schizopygopsis younghusbandi*, growth, exploitation rate, overfishing, fishery management, Yarlung Zangbo River

## Abstract

*Schizopygopsis younghusbandi*, a species widely distributed in the Yarlung Zangbo River Basin, serves as a crucial indicator of aquatic ecosystem stability. However, research on the current status of its populations in key tributaries of the basin remains limited. In this study, we employed biological modeling to assess the populations of *Schizopygopsis younghusbandi* in the Duoxiong Zangbo, Lhasa, Niyang, and Nianchu rivers. Our findings reveal that the populations in the Lhasa, Nyang, and Nyanchu rivers are overexploited. In light of these findings, we propose targeted conservation measures to restore and protect *Schizopygopsis younghusbandi* populations.

## 1. Introduction

The Tibetan Plateau, renowned as the “Roof of the World” and the “Asia Water Tower” [1,2], represents one of the highest-elevation and most geomorphologically complex plateaus globally [3,4]. The Yarlung Zangbo River, traversing the southern margin of the Tibetan Plateau, constitutes a critical fluvial system [5,6]. Its mainstream and tributaries together form a complex network of water systems [7], with unique geoenvironmental and climatic conditions fostering abundant plateau aquatic life [8,9,10]. Compared with freshwater fishes in plains, the Schizothoracinae exhibit distinct biological characteristics, including slower growth rate [11,12], delayed sexual maturation [13,14,15,16], and reduced fecundity [14,15,16,17]. The structure simplistic aquatic ecosystem of the plateau area demonstrates heightened environmental sensitivity [18]. Notably, as one of the largest ecologically sensitive areas on Earth [19,20], the Yarlung Zangbo River basin has a unique and fragile environment [21,22,23]. This susceptibility to climatic variability and anthropogenic impacts [4,23,24] renders Schizothoracinae populations particularly vulnerable to depletion with limited recovery capacity following ecological disturbances [25].

Tributaries constitute vital components of waters, serving as the main source of its recharge and playing a crucial role in maintaining basin-wide ecological equilibrium [3]. Among the numerous tributaries of the Yarlung Zangbo River, the Duoxiong Zangbo, Lhasa River, Niyang River, and Nianchu River have attracted particular scientific attention due to their unique hydrological characteristics and ecological functions. These alpine tributaries traversing high-altitude gorges and valleys provide optimal habitats for Schizothoracinae through their cold thermal regimes, rapid flow velocities, and elevated dissolved oxygen concentrations. Recent accelerated economic development in Tibet has intensified anthropogenic pressures [9], water conservancy facility construction, and overfishing, and invasive alien species have become increasingly prominent [26]. The inherent vulnerability of tributary ecosystems, characterized by low environmental carrying capacity and heightened sensitivity to anthropogenic disturbances, has exacerbated ecological impacts on aquatic biota—particularly Schizothoracinae populations [16]. Therefore, there is an urgent need to protect fishery resources in the Yarlung Zangbo River tributaries and establish effective management measures to ensure sustainable resources utilization [16,27].

Age and growth are critical components in research contents of fish individual biology, which not only reflect their growth characteristics but also are the fundamental indicators to evaluate population dynamics and resource sustainability [28]. These parameters hold significant implications for predicting fishery resource fluctuations in natural aquatic ecosystems. Concurrently, mortality parameters are a key indicator for assessing fishery resources and predicting the population trend [29], especially in the context of the dual effects of anthropogenic activities and climate change, where mortality shifts may directly threaten population stability.

*S. younghusbandi* belongs to the order Cypriniformes, subfamily Schizothoracinae, and genus *Schizopygopsis*, which are distributed in the Yarlung Zangbo River basin [30]. As an ecologically valuable fish resource [31], it serves as a key indicator species within this aquatic ecosystem. Existing research on *S. younghusbandi* primarily focuses on age-growth patterns [9,32], reproductive biology [33], spawning grounds [34], genetic diversity [26,31,35], and swimming ability [36], with all current studies concentrated on the mainstream of the Yarlung Zangbo River. However, the tributary waters of *S. younghusbandi* have not been reported on individual biology and population resources. In this context, this study investigates four principal tributaries of the Yarlung Zangbo River (Duoxiong Zangbo, Lhasa River, Niyang River, and Nianchu River) to analyze the age structure, growth characteristics, and resource development status of *S. younghusbandi*. Aiming to reveal the growth characteristics of the *S. younghusbandi* population, we constructed a population dynamics model, evaluating natural and anthropogenic impacts on mortality rates. The findings will provide a scientific basis and theoretical guidance for the conservation and management of *S. younghusbandi* wild resources. This study holds significant ecological and societal value by addressing research gaps in plateau fish ecology and providing crucial data for their evolution and germplasm conservation amid global warming and increasing human activities.

## 2. Materials and Methods

### 2.1. Study Area

This study focused on four tributaries of the Yarlung Zangbo River Basin, ordered by descending elevation: Duoxiong Zangbo (DX), Nianchu River (NC), Lhasa River (LS), and Niyang River (NY). Twenty sampling sites were established across these tributaries, with 6 sites in Duoxiong Zangbo, with 5 sites each in Niyang River and Lhasa River, and 4 sites in Nianchu River (Figure 1).

### 2.2. Sample Collection and Processing

Approved by the Tibet Autonomous Region fisheries authority, *S. younghusbandi* was collected in four tributaries from 2023 to 2024 using fishing nets such as gillnets and ground cages. Standardized sampling protocol involved setting 8 gillnets (two each of 2 cm, 3 cm, 4 cm, 5 cm each) and 2 ground cages per site, with gear deployed for 12 h (from 6 p.m. to 6 a.m. the next day). Captured specimens underwent biological measurements including total length (±1 mm), body length (±1 mm), and body weight (±0.1 g). Following dissection for gender determination, the lapillus was extracted, cleaned of connective tissue in alcohol, and preserved in 1.5 ml centrifuge tubes.

### 2.3. Lapillus Treatment and Age Determination

We secured the lapillus on the glass slide using colorless nail polish and allowed them to air-dry naturally. Sequential polishing using 800-grit, 1000-grit, and 3000-grit was conducted under stereomicroscope (Leica, Wetzlar, Gemany) monitoring until central growth rings became visible. When necessary, acetone was applied to flip the lapillus for optimal surface exposure. In the identification of lapillus growth rings, the combination of dark and translucent bands could be seen under incident light, and dark bands were used to record the age. The lapillus was observed with a stereomicroscope, counted, and photographed.

### 2.4. Data Statistics and Analysis Methods

#### 2.4.1. Body Length and Weight Relationship

The power function model was used to perform regression analysis of the relationship between body length and body weight of *S. younghusbandi* [37], and the body length and body weight data were analyzed using analysis of covariance (ANCOVA) to test whether there were significant differences in the length–weight relationships among different tributaries. The calculation formula was as follows:*W* = a *L^b^*
where *W* represents the body weight (g), *L* represents the body length (mm), and *a* and *b* are constant. The value of *b* can be used to determine whether the growth is uniform. When *b* is significantly different from from 3, the growth is allometric, whereas when it is close to or equal to 3, the growth is uniform.

#### 2.4.2. Growth Equation, Growth Rate Equation, and Growth Acceleration Equation

The Von Bertalanffy growth equation was applied to model the body length growth relationship of *S. younghusbandi* [38]. Subsequently, the body weight growth equation was derived based on body length and weight relationship analysis. Analysis of covariance (ANCOVA) was employed to test significant differences in length–age and weight–age relationships among individuals from four tributaries. First and second derivatives of both length- and weight-growth equations were calculated to establish corresponding growth velocity and acceleration equations. Key biological parameters were determined including the inflection point age, critical age, and growth performance index. The calculation formulas for these parameters are as follows:
Body length growth equation: *L*_t_ = *L*_∞_ (1 − e^−*k*(*t*−*t*_0_)^);Body weight growth equation: *W_t_* = *W*_∞_ (1 − *e*^−*k*(*t*−*t*_0_)^);Body length growth rate equation: *dL*/*dt* = *L*_∞_
*k*
*e*^−*k*(*t*−*t*^_0_^)^;Body weight growth rate equation: *dW*/*dt* = *b*
*W*_∞_
*k*
*e*^−*k*(*t*−*t*_0_)^ (1 − *e*^−*k*(*t*−*t*_0_)^)^*b*−1^;Body length growth acceleration equation: *d*^2^*L*/*dt*^2^ = −*L*_∞ _
*k*^2^*e*^−*k*(*t*−*t*_0_)^;Body weight Growth Acceleration Equation: *d*^2^*W*/*dt*^2^ = *bW*_∞_*k*^2^*e* − *k* (*t* − *t*_0_) (1 − *e*^−*k*(*t−t*_0_)^)^*b*−2^ (*be*^−*k*(*t*−*t*_0_)^ − 1);Inflection point age: *t_i_* = ln *b*/*k* + *t*_0_;Critical age: *t_c_* = [*Kt*_0_ − ln*M* + ln (*bK* + *M*)]/*K*;Growth performance index: φ = lg*k* + 2 lg*L*_∞_.
where *t* represents the age; *t*_0_, *t_i_* and *t_c_* represent the hypothetical theoretical starting age, inflection point age, and critical age, respectively; *L_t_* and *W_t_* represent the body length (mm) and weight (g) at age *t*; *L*_∞_ represents the asymptotic body length (mm); *W*_∞_ represents the asymptotic body weight (g); and *k* represents the growth coefficient.

#### 2.4.3. Mortality Characteristics and Exploitation Rate

The total mortality coefficient (*Z*) was calculated using the formula proposed by Beverton–Holt. Natural mortality coefficient (*M*) was estimated through Pauly’s formula [39,40,41]. The fishing mortality coefficient (*F*) was derived as the total mortality coefficient (*Z*) minus the natural mortality coefficient (*M*). The population exploitation rate (*E*) was the ratio of the fishing mortality coefficient (*F*) to the total mortality coefficient (*Z*), as follows:*Z* = *K* (*L*_∞_ − *L_mean_*)/(*L_mean_* − *L_c_*)ln*M* = −0.0066 − 0.279 ln*L*_∞_ + 0.6543 ln*K* + 0.4634 ln*T**F* = *Z* − *M**E* = *F*/*Z*
where *K* represents the growth rate, *L*_∞_ represents the asymptotic body length (mm), *L_mean_* represents the average body length of the sample (mm), *L_c_* represents the starting body length (mm), and *T* represents the annual average water temperature (°C) of the habitat water body for the studied fish species.

#### 2.4.4. Relative Units Replenish Catches and Biomass

The Beverton–Holt dynamic integrated model was used to evaluate the resource utilization of *S. younghusbandi*, employing the relative unit replenishment catches (*Y*′*/R*) and relative unit replenishment biomass (*B*′*/R*) curves. *E*-max represents the exploitation rate at which maximum production is achieved; E-10 corresponds to the exploitation rate when the marginal yield of the resource group decreases to 1/10 of that of the theoretical original resource and is regarded as the optimal exploitation rate. E-50 indicates the exploitation rate at which the resource biomass is reduced to 50% of its original level. The formula for calculating the correlation model [40] is*Y*′/*R* = *E* × (1 − *L_c_*/*L*_∞_) ^M/*k*^ [1 − 3(1 − *L_c_*/*L*_∞_)/(1 + *k/*Z) + 3 (1 − *L_c_*/*L*_∞_)^2^/(1 + 2 *k/*Z) − (1 − *L_c_*/*L*_∞_)^3^/(1 + 3 *k/*Z)]*B*′/*R* = (*Y*′/*R*)/(Z − M)
where *E* represents the exploitation rate; *L*_∞_ (mm) represents the asymptotic body length; *L_c_* (mm) represents the length of the starting body; *M* represents the natural mortality factor; *Z* represents the total mortality coefficient; and *k* represents the growth factor.

## 3. Results

### 3.1. Group Structure

This survey collected a total of 2058 *S. younghusbandi* specimens from four tributaries: 499 from Duoxiong Zangbo, 814 from Lhasa River, 262 from Niyang River, and 483 from Nianchu River. The body length of specimens across the four tributaries ranged from 20 to 412 mm, with body weights spanning 0.3~859.7 g. The female-to-male ratios were 0.78:1 (Duoxiong Zangbo), 1.38:1 (Lhasa River), 1.49:1 (Niyang River), and 1.34:1 (Nianchu River). The specimens showed an age range of 0−13 years (14 age compositions in total). Duoxiong Zangbo exhibited relatively uniform age distribution, while Lhasa River, Niyang River, and Nianchu River all demonstrated dominant age groups concentrated in the 0–3-year range (Table 1, Figure 2).

### 3.2. Body Length and Weight Relationship

Analysis of covariance (ANCOVA) revealed significant differences in the length–weight relationships of *S. younghusbandi* across tributaries (*F* = 19.298, *p* < 0.05). Consequently, separate length–weight relationships were established for each tributary (Figure 3). All four tributaries showed b-values approximating 3, indicating isometric growth patterns in *S. younghusbandi* throughout four tributaries.

### 3.3. Growth Equation, Growth Velocity Equation, and Growth Acceleration Equation

The measured data were used to fit the Von Bertalanffy growth equations of individual length and weight of *S. younghusbandi* from four tributaries. The results of covariance analysis (ANCOVA) revealed significant differences in length–age (*F* = 15.085, *p* < 0.05) and weight–age (*F* = 29.081, *p* < 0.05) among tributaries, necessitating separate Von Bertalanffy equations for each tributary (Figure 4 and Figure 5, Table 2).

In this study, the van Bertalanffy growth equation was used to describe the growth characteristics of *S. younghusbandi*. Growth curves across all tributaries exhibited identical trends under the Von Bertalanffy model. While length growth rate and acceleration showed monotonous decline with age (no inflection points), weight growth demonstrated distinct phases: accelerated growth prior to the inflection age followed by progressive deceleration thereafter. In addition, the growth inflection point age, critical age, and growth performance index of the four tributaries were obtained (Table 3). Both inflection and critical age were basically decreased with descending river elevation, whereas growth performance index showed minimal variation among the four tributaries.

### 3.4. Mortality Characteristics and Exploitation Rate

The results of the mortality and exploitation rates of *S. younghusbandi* in the four tributaries are summarized in Table 4. Exploitation rates (*E*) ranged from 0.313 to 0.777 across tributaries, with Niyang River exhibiting the highest exploitation rate (0.777), followed by Lhasa River (0.659), Nianchu River (0.522), and Duoxiong Zangbo (0.313). Among them, the *E* of Lhasa, Niyang, and Nianchu rivers were higher than their respective *E-*max, while only Duoxiong Zangbo was lower than the *E-*max (Table 4).

### 3.5. Relative Units Replenish Catches and Biomass

The relationship between exploitation rate (*E*) and both yield per recruit (*Y*′*/R*) and the biomass per recruit (*B*′*/R*) of *S. younghusbandi* from four tributaries are illustrated in Figure 6. When exploitation rate was reduced to *E*-max, that is, 0.557 (Duoxiong Zangbo), 0.448 (Nianchu River), 0.617 (Lhasa River), and 0.616 (Niyang River), the corresponding *Y*′*/R* reached the maximum. When the exploitation rate decreased to *E*-50, all tributaries showed minor *Y*′*/R* declines. Concurrently, biomass retention dropped below 25% at *E*-max but stabilized at 50% when exploitation rate reached *E*-50 (Figure 6).

Analysis of *Y*′*/R*, *E* and *L*_c_ isopleths for *S. younghusbandi* in four tributaries (Figure 7) revealed that if the current *E* remained constant, *Y*′*/R* would increase with the increasing *L*_c_. When *L*_c_/*L*_∞_ = 0.6 across tributaries, the corresponding *L*_c_ values maximizing of *Y*′*/R* were 248 mm (Duoxiong Zangbo), 240 mm (Nianchu River), 233 mm (Lhasa River), and 236 mm (Niyang River).

## 4. Discussion

### 4.1. Selection of Age Identification Materials and Growth Models for S. younghusbandi

Accurate identification of fish age is fundamental for the studying of population growth characteristics and resource assessment. Current methodologies predominantly rely on annuli patterns retained on calcified structures such as scales and lapillus, but the most suitable identification materials for age determination vary across species. Compared to scales that are susceptible to abrasion-induced annuli loss, lapillus is known for its stable annulus structure and anti-reabsorption properties [9,42,43], showing significant advantages in the age determination of long-lived fish [44]. The morphological characteristics of lapillus in this study revealed distinctive alternations between hyaline zones and dark bands with clearly discernible annuli. The successful identification of a 14-year-old specimen confirms lapillus as the optimal age identification material for this species, consistent with established findings in Schizothoracinae [9].

Regarding growth model selection, this study evaluated three classical growth equations. The Gompertz equation, constrained by its short-lifecycle assumption [45], proved inadequate for modeling the long-lived *S. younghusbandi*. The Logistic equation demonstrated sensitivity to environmental heterogeneity—a critical limitation given the notable habitat variations across four Yarlung Zangbo River tributaries where specimens were collected [46]. In contrast, the Von Bertalanffy growth equation, based on metabolic theory [47,48], provided better modeling accuracy for the asymptotic growth pattern of *S. younghusbandi*. Therefore, the Van Bertalanffy equation is the optional growth equation for *S. younghusbandi* in this study.

### 4.2. Growth Characteristics of Lhasa Bare Schizori

Fish growth is driven by both genetic and environmental factors, with environmental variation being the key driver of growth differentiation within the same species [49]. Growth parameters serve not only as the foundation for studying the biological and ecological characteristics of fishes, but also as crucial elements in constructing fishery resource assessment models [50,51,52].

The growth coefficient *k* is one of the most critical parameters to describe the growth characteristics of fish [53,54]; based on the theoretical framework of the growth coefficient (*k*), we revealed the ecological adaptation characteristics of *S. younghusbandi* across four Yarlung Zangbo River tributaries. The results indicate that tributary populations exhibited a *k*-value of 0.154–0.174, aligning with the steady growth pattern (0.10–0.20). Longitudinal comparison revealed significantly higher growth rates in the tributary population compared to the mainstream population in 2002–2005 (*k*_male_ = 0.0738; *k*_female_ = 0.0789) [32], but lower than the 2008–2009 mainstream population (*k*_male_ = 0.194; *k*_female_ = 0.233) [9]. This temporal variation stems from two drivers—habitat environmental change [55] and excessive fishing pressure [56]: (1) global warming has accelerated fish growth due to rising water temperatures [24,57], though tributaries maintain cooler temperature with altitudinal gradients (higher elevation populations show lower *k* value). (2) Persistent high-intensity fishing pressure drives phenotypic plasticity-mediated life history adjustments [58]—compensatory adaptation through accelerated growth rates sustains population persistence [59,60]. This mechanism concurrently accounts for observed growth rate escalation in mainstream populations [9,32]. Notably, elevated growth rates in tributary populations signal intense fishing pressure [61], manifested through altered life history parameters: accelerated growth, as well as earlier inflection ages accompanied by the decrease in the maximum asymptotic body length, indicating that the population structure tends to be younger and smaller.

In this study, population structure variation and the driving mechanism of *S. younghusbandi* inhabiting tributaries of the Yarlung Zangbo River were evaluated by asymptotic body length (*L*_∞_) and growth pattern (*b*). The results demonstrated a significantly lower *L*_∞_ value in tributary populations (387.9–414.5 mm) compared to 2002–2005 mainstream records (*L*_∞male_ = 442.7 mm). *L*_∞female_ = 471.4 mm) [32], confirming population miniaturization with spatial differentiation: Lhasa River and Niyang River populations exhibited higher growth rates (*k*) and lower *L*_∞_, displaying marked juvenilization. Despite the lowest k value (0.154), Duoxiong Zangbo maintained higher *L*_∞_ (414.5 mm), indicating that its population structure was relatively stable. Growth pattern analysis revealed that only the *b* value of the Niyang River was >3 (body weight growth was dominant), and the *b* value of the other tributaries was <3 (body length growth was dominant). The occurrence of this difference is closely related to the thermodynamic gradient of the watershed [62,63]. The Niyang River, as a low-altitude water area in the middle reaches [64], has a high average annual temperature [65,66], plankton and other forage organisms were abundant [20], and their eutrophication habitats promoted fish energy accumulation and made fish plumpness higher in this area [63]. Conversely, upstream tributaries (Duoxiong Zangbo and Nianchu River) under cold stress [66,67] prioritize metabolic allocation to length growth, producing length-dominant growth trajectories.

The above analysis showed that the spatial differences of growth parameters among tributaries were regulated by two factors: (1) natural habitat heterogeneity—Duoxiong Zangbo and Nianchu rivers (upper Yarlung Zangbo River) exhibit high elevation [66], constrained channels with narrow surface [67], and low water temperature. Conversely, the Lhasa River and the Niyang River (middle reaches) feature gradual banks [64,68], expansive water surface, and high water temperature [65,66,69], with abundant forage resources supporting fish development [20,63]. (2) Anthropogenic disturbance intensity—there are no large towns on either side of the Duoxiong Zangbo, there are basically no fishing activities along the river, and no water storage dams are currently being built, which maintains the natural hydrological connectivity. Conversely, Nianchu River, Lhasa River, and Niyang River drain through Shigatse, Lhasa, and Nyingchi metropolitan areas, respectively [64,67,68], experiencing compounded stressors: intensive fishery activities, invasive exotic species [63], and cascade hydropower station development, significantly impacting fish growth and reproduction. These cumulative stressors drive *S. younghusbandi* to enact phenotypic plasticity-mediated life history adjustment [58], potentially inducing systematic growth parameter shifts.

The growth coefficient value may be affected by population age structure. The growth characteristic index (φ) combines the performance of k and *L*_∞_ well, so we employed φ to verify the reliability of the above growth parameter estimation [70] and to evaluate the intraspecific growth heterogeneity of *S. younghusbandi*. Data show that φ of the four tributaries converge at 4.41–4.43, and their narrow fluctuations confirm the robustness of the *K*-*L*_∞_ synergy. Cross-population comparisons showed that φ in this study were in the median range of φ of Schizothoracinae on the Tibetan Plateau (4.37–4.75) [47,71,72,73,74,75,76,77]. These findings indicate that the growth performance of *S. younghusbandi* in four tributaries was at a moderate level, and it is implied that its growth strategy was phylogenetically constrained and did not break through the energy allocation pattern of its cozoar relatives. These results further support the conclusion: inter-tributary growth parameter differences primarily stem from phenotypic plasticity responses, not genetic adaptation, which provides a quantitative basis for formulating basin-specific resource restoration plans.

### 4.3. Current Status and Protection Measures of Bare Rift Resources in Lhasa

Fish mortality constitutes a pivotal driver of population dynamics, and its rate determines the degree of population decline. Based on the reliability criterion of the growth equation (*e*^−k^ < 1) [78], the reliability range of natural mortality (*M/K* = 1~2.5), and the discrimination threshold of death type (the relationship between Z/K and 3) [78], it was found that the *e*^−k^ (0.840–0.857) and *M/K* (2.02–2.17) of the four tributaries of *S. younghusbandi* were within the range of the theoretical standard. The results indicate that the growth equation is reliable, and the natural mortality estimate is highly reliable. The *Z/K* values showed that only the death category of Duoxiong Zangbo (*Z/K* < 3) was dominated by natural causes, whereas the main causes of death in the Nianchu River, Lhasa River, and Niyang River (*Z/K* > 3) were anthropogenic fishing, with the results of this study revealing that the latter three tributaries were overfished, and the situation was more serious.

Exploitation rate analysis revealed distinct patterns: *S. younghusbandi* in Duoxiong Zangbo maintains an exploitation rate (*E* = 0.313) below both Gulland’s threshold (0.5) [79] and Mehanna’s river-specific safety range (*E* < *E*-max) [80]. In contrast, the Nianchu River (*E* = 0.522), Lhasa River (*E* = 0.659), and Niyang River (*E* = 0.777) exhibit overexploitation, exceeding both thresholds. The above results indicate that the population exploitation rate of Duoxiong Zangbo remains low (within safe thresholds) with a relatively intact population structure, whereas the other three tributaries exhibit overexploitation patterns. Notably, rapid regional economic development and surging market demands in the Lhasa River and Niyang River basins have intensified fishing activities [81], leading to significant degradation of their germplasm resources.

The current development and utilization of *S. younghusbandi* resources in the Nianchu River, Lhasa River, and Niyang River basins exhibits irrational exploitation patterns, and resource management can be optimized by increasing the fishing body length and reducing the fishing intensity [82]. The Beverton–Holt dynamic model analysis demonstrates that adjusting catchable length proves more effective than simply reducing fishing intensity in balancing resource conservation and utilization efficiency [83]. In this study, it is suggested that the three indexes’ average of the maximum catch per unit recruitment in the Beverton–Holt dynamic model, inflection point age-specific length, and critical age body length should be used as the optimal minimum catchable length. Accordingly, the recommended lengths are 257 mm (Duoxiong Zangbo), 249 mm (Nianchu River), 237 mm (Lhasa River), and 242 mm (Niyang River). This strategy facilitates increased individual recruitment into the *S. younghusbandi* population [82], maximizes its growth potential, avoids the occurrence of recruitment overfishing, and ensures the availability and sustainability of the resource [83].

Based on the integrated research findings of the growth ecology of *S. younghusbandi* and the vulnerability of aquatic ecosystem in the Tibetan Plateau, the following conservation strategies are proposed: (1) The scientific regulation mechanism of fishery resources. Establish differentiated minimum catchable length standards based on population dynamics models [82], implement catch quota systems and seasonal fishing moratoria [84,85,86,87], strengthen fisheries monitoring systems [88] to combat illegal fishing, and achieve sustainable balance between exploitation rates and population recruitment through catch statistics and age–structure analyses [88]. (2) Habitat integrity restoration projects [89]. In view of the habitat fragmentation caused by the development of river cascade hydropower stations [88], priority was given to the construction of natural fish passage channels at key migration nodes to ensure the reproductive migration of fish. (3) Biological invasion prevention and control systems [90]. Establish a risk assessment model for the invasion species, strictly control regulate fish stocking practices, and safeguard niche breadth and trophic overlap indices of native fish populations.

## 5. Conclusions

In this study, we examined the population structure and resource status of *S. younghusbandi* in four tributaries of the Yarlung Zangbo River Basin: the Duoxiong Zangbo, Nianchu, Lhasa, and Niyang rivers. The results indicated that while the exploitation rate of *S. younghusbandi* in the Duoxiong Zangbo remained within sustainable limits, populations in the Nianchu, Lhasa, and Niyang rivers were subject to overexploitation. The Beverton–Holt dynamic pool model further revealed high fishing intensity and small size at first capture across all four tributaries. Based on these findings, we recommend implementing minimum catch sizes of 257 mm (Duoxiong Zangbo), 249 mm (Nianchu River), 237 mm (Lhasa River), and 242 mm (Niyang River) for *S. younghusbandi*. These results provide valuable insights into developing conservation strategies and standardized management practices for this species in the studied tributaries.

## Figures and Tables

**Figure 1 biology-14-00707-f001:**
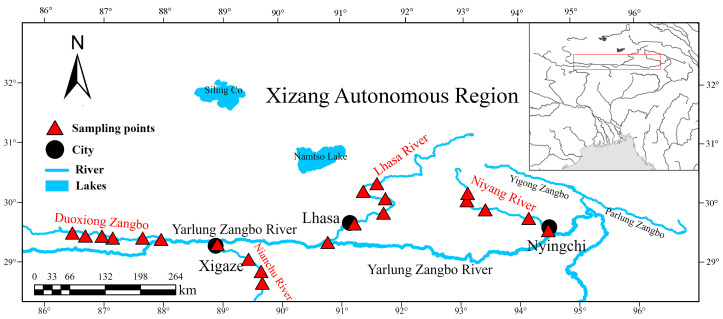
Sampling site.

**Figure 2 biology-14-00707-f002:**
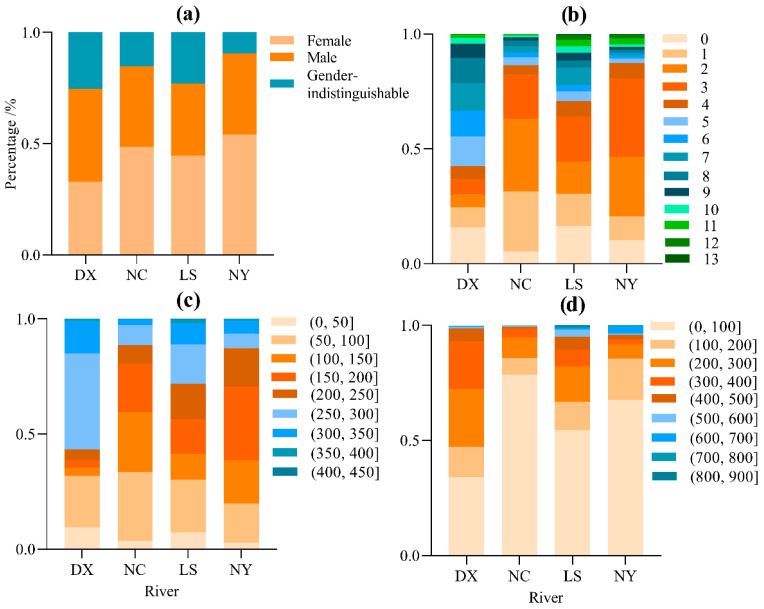
Population structure of four tributaries of *S. younghusbandi.* Note: (**a**) sex proportion; (**b**) age proportion; (**c**) proportion of the body length group; (**d**) proportion of the body weight group.

**Figure 3 biology-14-00707-f003:**
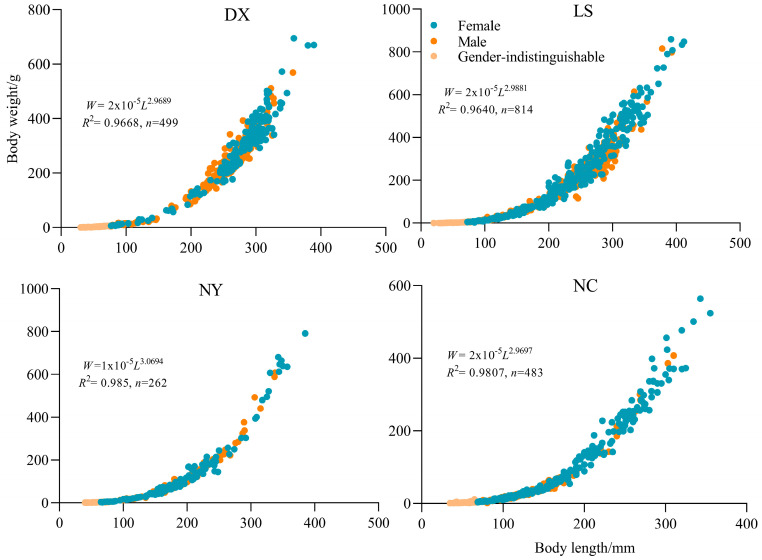
Relationship between body length and body weight of *S. younghusbandi*.

**Figure 4 biology-14-00707-f004:**
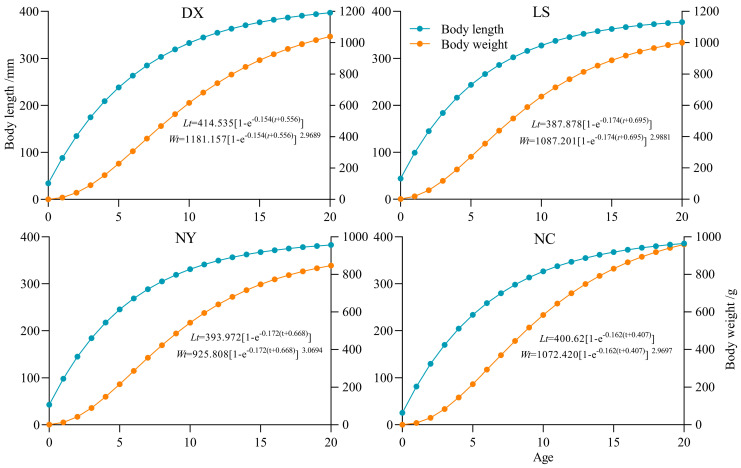
Growth equation of four tributaries of *S. younghusbandi*.

**Figure 5 biology-14-00707-f005:**
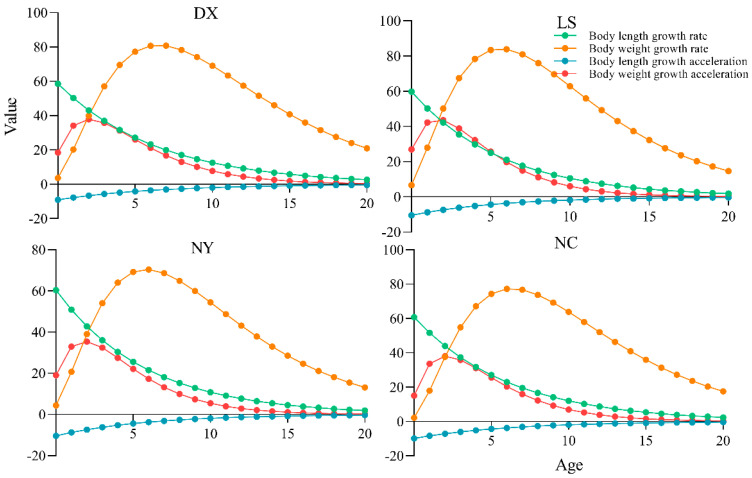
Growth rate and growth acceleration of four tributaries of *S. younghusbandi*.

**Figure 6 biology-14-00707-f006:**
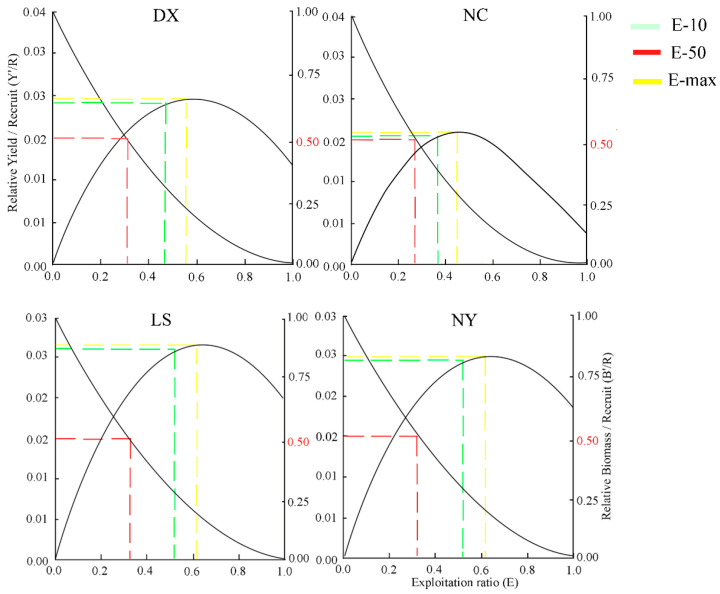
Two-dimensional analysis of *Y*′/*R* and *B*′/*R* of four tributaries of the *S. younghusbandi*.

**Figure 7 biology-14-00707-f007:**
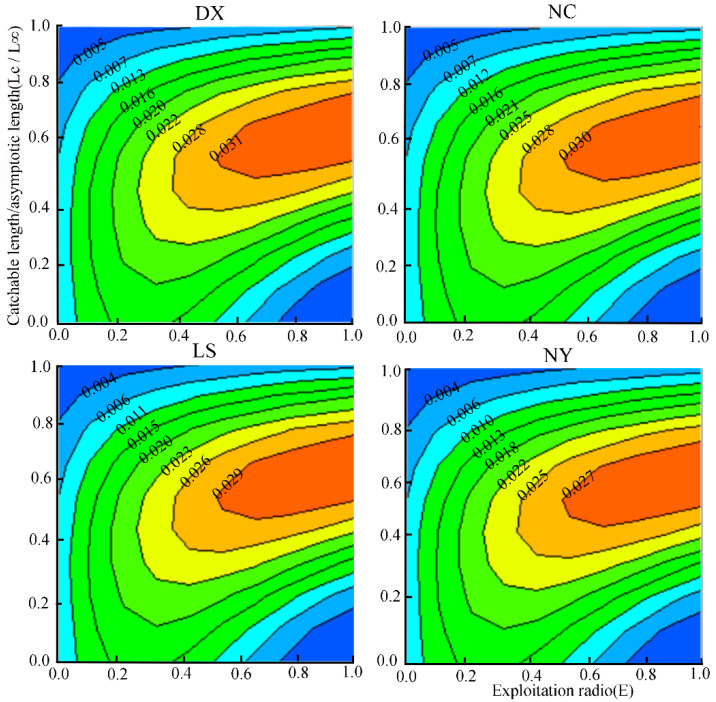
Relative yield-per-recruit (*Y*′/*R*) of four tributaries of the *S. younghusbandi* in relation to exploitation rate and catchable length.

**Table 1 biology-14-00707-t001:** The body length and body weight of *S. younghusbandi*.

River	Number	Number		Body Length/mm		Body Weight/g	
		Female	Male	Range	Mean	Range	Mean
Duoxiong Zangbo	499	164	209	30~389	205.89	0.3~695.2	190.63
Nianchu River	483	235	175	35~355	142.07	0.6~563.6	73.23
Lhasa River	814	364	264	20~412	179.24	0.3~859.7	151.47
Niyang River	262	142	95	40~385	168.24	1.0~791.0	109.89

**Table 2 biology-14-00707-t002:** Growth rate and growth acceleration equation of four tributaries of *S. younghusbandi*.

Index	River	Equation
Body length growth rate	Duoxiong Zangbo	*dL*/*dt* = 63.838 e^−0.154(*t*+0.556)^
	Lhasa River	*dL*/*dt* = 67.491 e^−0.174(*t*+0.695)^
	Niyang River	*dL*/*dt* = 67.763 e^−0.172(*t*+0.668)^
	Nianchu River	*dL*/*dt* = 64.900 e^−0.162(*t*+0.407)^
Body length growth acceleration	Duoxiong Zangbo	*d*^2^*L/dt*^2^ = −9.831 e^−0.154(*t*+0.556)^
	Lhasa River	*d*^2^*L/dt*^2^ = −11.743 e^−0.174(*t*+0.695)^
	Niyang River	*d*^2^*L/dt*^2^ = −11.655 e^−0.172(*t*+0.668)^
	Nianchu River	*d*^2^*L/dt*^2^ = −10.514 e^−0.162(*t*+0.407)^
Body weight growth rate	Duoxiong Zangbo	*dW*/*dt* = 540.037 e^−0.154(*t+*0.556)^ [1 − e^−0.154(*t+*0.556)^]^1.9689^
	Lhasa River	*dW*/*dt* = 565.268 e^−0.174(*t+*0.695)^ [1 − e^−0.174(*t+*0.695)^]^1.9881^
	Niyang River	*dW*/*dt* = 488.768 e^−0.172(*t+*0.668)^ [1 − e^−0.172(*t+*0.668)^]^2.0694^
	Nianchu River	*dW*/*dt* = 515.932 e^−0.162(*t+*0.407)^ [1 − e^−0.162(*t+*0.407)^]^1.9697^
Body weight growth acceleration	Duoxiong Zangbo	*d*^2^*W/dt*^2^ = 83.166 e^−0.154(*t*+0.556)^ [1 − e^−0.145(*t*+0.556)^]^0.9689^ [2.9689 e^−0.154(*t*+0.556)^ − 1]
	Lhasa River	*d*^2^*W/dt*^2^ = 98.357 e^−0.174(*t*+0.695)^ [1 − e^−0.174(*t*+0.695)^]^0.9881^ [2.9881 e^−0.174(*t*+0.695)^ − 1]
	Niyang River	*d*^2^*W/dt*^2^ = 85.134 e^−0.172(*t*+0.668)^ [1 − e^−0.172(*t*+0.668)^]^1.0714^ [3.0714 e^−0.172(*t*+0.668)^ − 1]
	Nianchu River	*d*^2^*W/dt*^2^ = 83.581 e^−0.162(*t*+0.407)^ [1 − e^−0.162(*t*+0.407)^]^0.9697^ [2.9697 e^−0.162(*t*+0.407)^ − 1]

**Table 3 biology-14-00707-t003:** The inflection point age, critical age, and growth performance index of the four tributaries of *S.younghusbandi*.

River	Inflection Point Age	Critical Age	Growth Performance Index
Duoxiong Zangbo	6.51	5.32	4.42
Nianchu River	6.31	5.13	4.41
Lhasa River	5.60	4.40	4.42
Niyang River	5.85	4.46	4.43

**Table 4 biology-14-00707-t004:** Mortality coefficient and exploitation rate of four tributaries of the *S. younghusbandi*.

River	Total Mortality (*Z*)	Natural Mortality (*M*)	Exploitation Rate (*E*)	*E*-Max
Duoxiong Zangbo	0.453	0.311	0.313	0.557
Lhasa River	1.066	0.364	0.659	0.617
Niyang River	1.67	0.373	0.777	0.616
Nianchu River	0.693	0.331	0.522	0.448

## Data Availability

The data that support the findings of this study are available from the corresponding author upon reasonable request.

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
