# Peer review of "Population Structure, Growth Characteristics, Resource Dynamics, and Management Strategies of Schizopygopsis younghusbandi in Four Tributaries of the Yarlung Zangbo River, Tibet"

_biology, 2025, doi:10.3390/biology14060707_

Round 1

Reviewer 1 Report

Comments and Suggestions for Authors

I believe this manuscript is very interesting, given the data it gives on the stock assessment of Schizopygopsis younghusbandi. The article is generally made with length, weight, and age structure data and written with appropriate data analysis. Nevertheless, it needs improvement in order to improve for better understanding. 

Comments on the Quality of English Language

The English could be improved to more clearly express the research.

Reviewer 2 Report

Comments and Suggestions for Authors

This paper is a solid, sound study of the growth of Schizopygopsis younghusbandi, a cyprinid (carp family) fish of Tibet. The paper concludes that the fish is clearly overfished in three of four major tributaries of the Yarlung Zangpo, the major river of Tibet. (Other papers cited in the bibliography report that the fish is in at least as bad shape in the Yarlung itself.) Overfishing, dams, and introduced fish all diminish its population. It grows well only in the Duoxiong River, a major tributary of the Yarlung but one far upstream and draining cold, difficult terrain, so there are few people there to fish or build dams. Duoxiong is routinely misspelled as "Dogxung" in the paper, which must be corrected. The other rivers have large cities situated on them, so are more prone to overfishing and other problems. Growth is slow in this species, but is faster in the Nyingchi River, which is lower in altitude and thus warmer and more productive. (It is also the home of the world's second tallest tree.) This fish grows slowly and rarely reaches more than 40 cm in length, but is an important food resource locally. The paper ends with pleas for conservation, protection, and sustainable fishing, with the usual recommendations for size limits, fish channels around hydroelectric projects, and control of introduced species. This paper is a straightforward data paper, reporting on a well-known fish, but it breaks important new ground in several ways: it reports on the status of the fish in several major rivers not previously studied in regard to this fish; it does a superbly thorough job of reporting growth, size, and related statistics; and it provides information on unsustainable exploitation, which is extremely important given that this is a major food fish resource.

The English is very good, but note a few minor misspellings in names ("Bertalan" for Bertalanffy at one point, "Bverton" for Beverton) and the major error in the river name ("Dogxung" for Duoxiong). 

----------

All good--no need to go into so much detail. They used standard methodology for measuring the size of their 2000+ sample fish, and used the von Bertalanffy growth modeling statistic that has been pretty much standard for decades. They use a Beverton-Holt model for resource use by fish. It too has been standard for decades. Thus there is not really much to say about the methods here except that they are thoroughly appropriate, quite standard, and used pretty much everywhere. The conclusion they reach is that the fish are suffering population reduction in 3 of the rivers, those with sizable cities on their banks. Overfishing, dams, and competition from introduced fish are all mentioned. This is already well documented for the main Yarlung Zangpo itself, so what these researchers have done is simply extend the study to four major Tibetan tributaries. The results are very important from a fisheries point of view, but not innovative or surprising scientifically--just good, solid, well-done work to show what is needed in conserving these fish. Their suggestions are also standard and perfectly reasonable and well taken. All in all a good study that shows where this species is overexploited and what we should do about it.
